# On the Production of Potassium Carbonate from Cocoa Pod Husks

**Kouwelton Kone [1]**, **Karl Akueson [2]** and **Graeme Norval [3,*]**

[1] DFR Génie Chimique et Agro-Alimentaire, Institut National Polytechnique Félix Houphouët-Boigny de Yamoussoukro, Yamoussoukro BP 1093, Cote D'Ivoire; kouwelton.kone@inphb.ci

[2] La Financière de l'Eléphant, 1er Etage, Immeuble Cormoran, Deux Plateaux Vallon, Cocody, Abidjan, Cote D'Ivoire; karl@lafinele.com

[3] Department of Chemical Engineering and Applied Chemistry, University of Toronto, Toronto, ON M5S 3E5, Canada

\* Correspondence: Graeme.norval@utoronto.ca

**Abstract:** Cocoa beans are found inside an outer husk; 60% of the cocoa fruit is the outer husk, which is a waste biomass. The husk cannot be used directly as a soil amendment as it promotes the fungal black pod disease, which reduces crop yield. The pods are segregated from the trees, and their plant nutrient value is wasted. This is particularly true for the small acreage farmers in West Africa. Cocoa pod husk is well suited to be used as a biomass source for electricity production. The waste ash is rich in potassium, which can be converted in various chemical products, most notably, high-purity potassium carbonate. This study reviews the information known about cocoa and cocoa pod husk, and considers the socio-economic implications of creating a local economy based on collecting the cocoa pod husk for electricity production, coupled with the processing of the waste ash into various products. The study demonstrates that the concept is feasible, and also identifies the local conditions required to create this sustainable economic process.

**Keywords:** potassium carbonate; cocoa pod husk; biomass ash

## 1. Introduction

Cocoa is native to central America, where it normally grows beneath the canopy of taller trees. The yield of cocoa is lower when cocoa is grown in this fashion. Clearing the shade trees increases the amount of direct sunlight reaching the cocoa trees, and the yield increases. The increase in yield is short-lived; the cocoa yield begins to be limited by soil conditions, and in particular, the potassium content of the soil [1]. A typical practice is to add fertilizers; these include the standard NPK (Nitrogen, Phosphorus, Potassium) mixtures, as well as calcium and magnesium supplements, depending on the soil conditions. Typical fertilization rates are in the range of 100–500 kg/annum per ha.

Cocoa beans are rich in inorganic matter; typical analyses are 2.5% K in the dry beans. At a cocoa production rate of 600 kg dry beans/ha, the potassium lost with the cocoa product is of the order of 15 kg K/ha [2]. When cocoa is harvested, the seeds are taken from the pods; 60% of the mass of the cocoa pod is the husk. Fresh cocoa pod husk contains ~1.6% K and 0.7% P [3]. The leaves contain a further ~1.5 wt% K.

In principle, the leaf litter and cocoa pod husk should be spread underneath the trees, in order to boost the organic content of the soil, as well as to provide K and P, as well as Ca and Mg. Black pod disease is a fungal rot of the cocoa pod, which reduces the cocoa yield, with yield losses of up to 30% [4]. As with other fungal diseases, affected pods must be segregated from the rest of the farm and destroyed, so that the spores are not able to infect subsequent crops.

If cocoa is grown as part of a mixed agricultural system, the soil nutrients could be replaced by the organic matter from the other trees [5]. Unfortunately, this reduces the acreage available for cocoa. Most cocoa farms are small, a few acres, and the farmers rely on increased cocoa yields [6].

It has long been known that the ash produced when cocoa pod husks are burned is rich in potash [7]. The ashes are leached, giving an alkaline solution, rich in potassium hydroxide and potassium carbonate. This solution has been used as the alkali source for local soap production.

Further, when the cocoa pod husks are burned, the black pod fungus is destroyed, greatly reducing the likelihood of future black pod outbreaks. It follows that the spoiled cocoa pods as well as the cocoa husks should be burned because ashing will improve the control of black pod disease, as well as provide a valuable mineral source.

A sustainable chemical business is presented in which the cocoa pod husks are burned, generating electricity in the remote farm regions, and the biomass ash is converted in a suite of chemical products for use. The issue of the potassium balance is considered, and techno-commercial considerations are presented.

## 2. Results

### 2.1. Analyses of the Husk and Ash

The cocoa pod husk had a higher heaving value of 18.1 MJ/kg (dry husk) and a lower heating value of 17.1 MJ/kg (dry husk), with a 12.7% ash content. The major constituents of the ash were K (320 g/kg), Ca (65 g/kg), Mg (42 g/kg), and Si (9.1 g/kg). The other elements were reported as S (7100 mg/kg), P (5600 mg/kg), Al (1100 mg/kg), and Mo (1100 mg/kg), followed by Ba (250 mg/kg), Cd (<3 mg/kg), Co (< 10 mg/kg), Cr (<10 mg/kg), Cu (150 mg/kg), Fe (940 mg/kg), Ni (33 mg/kg), Pb (<10 mg/kg), Ti (69 mg/kg), V (<25 mg/kg), and Zn (530 mg/kg).

### 2.2. The Farmer's Considerations

For the purpose of this analysis, a small cocoa farm (5 ha) will be considered. Farms of this size are family run; they tend to have older trees, and also have lower yields linked to the low use of chemical fertilizer [6]. For this farm, we can consider the balance loss to both the cocoa bean and the cocoa pod. For this calculation, the ratio of wet pod to dry beans is 10:1; 60% of the wet pod is wet husk, of which 1.6% is K and 0.7% is P.

The 5-ha farm generating 250 kg/ha of dry beans will produce 12.5 tons/annum of wet pod. After ashing, the farm would have 825 kg/annum of ash. This would be split into 500 kg/annum of potassium carbonate and 325 kg/annum of calcium/magnesium solid.

The farm would generate roughly 2500 $/annum through the sale of dry beans, at a price of 2000 US$ per ton. There would be an avoided cost of 620 $/annum, based on a fertilizer value of 750 US$/ton. The value of the ash is a significant improvement in the annual income of the farm.

## 3. Discussion

The ash contains the inorganic elements present in the cocoa husk. The predominant cations species are potassium, calcium, and magnesium (as well as sodium) and will be present as carbonates ($K_2CO_3$, $CaCO_3$, $MgCO_3$, $Na_2CO_3$). The phosphorus will be present as calcium phosphate, $Ca_3(PO_4)_2$.

Chemical analysis of cocoa pod husks from a variety of countries has been reported (3) and is consistent with these results. The average ash content was 10.7 wt%. The average potassium content was 1.63 wt%. This gives a $K_2CO_3$ yield of 5.76 wt% of the starting cocoa pod husks. The average calcium, magnesium, and phosphate contents are 0.33 wt%, 0.93 wt%, and 0.69 wt%. The insoluble calcium carbonate and magnesium hydroxide contents would be 0.82 wt% and 2.23 wt% of the starting cocoa pod husk.

The implication is that for every ton of cocoa pod husk, one will create ~60 kg of potassium carbonate and ~30 kg of a calcium/magnesium solid. The calcium/magnesium solid is best returned

to the fields. These are necessary nutrients, and working the ash under the cocoa trees is part of an effective fertilizer program. Indeed, this would replace the import of calcium-based fertilizers.

The raw ash is a mixture of inorganic carbonates, oxides, and phosphates. Given that these inorganic species originated in the soil, it would seem straightforward to ash the cocoa husk and cocoa wood, to prevent the spread of black pod and other fungal diseases, and then to spread the ashes under the trees. Two specific issues prevent the direct application of ashes onto the soil. The first is that the ashes are dry and powdery and tend to blow away when dry. More importantly, the ash is hydroscopic (it attracts water); indeed, the oxides react with water, giving a hydroxide, which is strongly alkaline. It is not pleasant to work with the hydroscopic and strongly alkaline ash powder.

The better approach is to add the ash to water, with agitation. The potassium (and sodium) salts dissolve readily, giving a solution of potassium carbonate and potassium hydroxide, which is strongly alkaline. The concentration of the solution depends on the ratio of ash to water, and the amount of potassium in the ash, limited only by solubility.

The potassium solution is an effective fertilizer that will replace the import of potassium-based fertilizers. The solution could be evaporated to dryness, giving a crystal product, or the solution fertilizer could be applied as is. Applying the potassium as solution is the lowest cost option.

The recovery of potassium carbonate from wood ashes has been a long history [8]. Indeed, the first patent granted in the United States covered an improvement in the process for the recovery of potassium carbonate from wood ashes [8]. The ash should be near white in color; poor burning leaves carbonaceous residue in the ash, giving it a red to brown to black color. Darker-colored ashes should be burned a second time, in order to remove the residual organic matter, followed by dissolution of the ash in water, giving a calcium/magnesium/phosphate containing solid, and a potassium-rich solution. The cocoa pod husk is an excellent source of potassium salts.

## 3.1. Individual Farm Analysis

The data in Table 1 lead to two conclusions. The first is that the sale of cocoa beans and the loss of the husk from the farms leads to soil depletion of potassium and phosphorus along with other inorganic minerals, such as calcium and magnesium. The biomass accumulates various inorganic elements, more of which are present in the husk than in the beans. Farms that have high yields also have higher losses of inorganic elements from the soil. This leads to depletion of these elements from the soil, thereby causing a decrease in crop yields over time. The data in the table also represents the fertilizer requirement (kg/a/ha) just to make up for the losses in the biomass; but, since soluble potassium and phosphate fertilizers are applied, additional fertilizer is required to make up for the material that dissolves into the groundwater, and escapes the farm. It is noted that there is a significant amount of potassium held in the cocoa wood, which is a further source of cocoa biomass.

**Table 1.** Potassium balance of a 5-ha farm.

| Yield (Dried Bean) | K (kg/a/ha in Dry Bean) | Wet Cocoa Pod Husk (kg/a) | K (kg/a/ha in Wet Husk) | P (kg/a/ha in Wet Husk) | K Loss (kg/a/ha) |
|---|---|---|---|---|---|
| 250 kg/ha | 6.25 | 1500 | 24 | 10.5 | 30.25 |
| 500 kg/ha | 12.5 | 3000 | 48 | 21 | 60.5 |
| 750 kg/ha | 18.75 | 4500 | 72 | 31.5 | 90.75 |

The mineral loss values in Table 1 are consistent with those of other reports [8]. Further, the recommended fertilizer additions [9] are significantly higher than the estimated loss of mineral; the recommended dosage rates are upwards of 350 kg/ha of K, based on a plantation of 1075 trees/ha. The dosage rate considers the K that is lost with the beans, and with the husk, as well as the potassium that accumulates in the growing wood; further, it recognizes that not all of the granular inorganic potassium ends up in the cocoa: A portion dissolves and drains into the ground water.

Finally, we note that it has long been known that wood ashes are rich in potassium; indeed, potash production was always located where forests were being cleared [7,8]. Cocoa plantations often have mixtures of other crops, such as coconut and plantain for shade. When these trees are harvested and burned, the ash will be similar in composition to that derived from the cocoa pod husk [10]. The ash can be mixed with that from the cocoa pod husk, thereby supplementing the yield of the potassium solution. Plantain is also rich in potassium, and the peelings and wood can be ashed with the ash being applied as a soil supplement.

As the farm returns the mineral content to the soil, there will be an increase in the yield of cocoa; the revenues and the value of the avoided costs both increase. This is especially true for the farm that does not use fertilizer supplements: The use of the cocoa pod husk ash allows them to increase the yield, using materials that they have on hand.

Additionally, many of the remote farming villages are not connected to the national electrical grid as they are too remote. The installation of a biomass to electricity plant will allow for the farming community to be electrified, leading to many subsequent improvements in living conditions.

## 3.2. Distribution of Ashing Sites

For the 5-ha farm, producing 250 kg/ha dry beans, the mass of dry pod is ~1875 kg. The 5 ha covers an area of 0.05 km$^2$. A circular radius of 1 km has 79 ha, or roughly 14 individual farms, and a dry pod yield of 29.6 ton.

The lower heat of combustion of dry cocoa pod husk is ~17 MJ/kg. The cocoa pod husk would be collected at harvest, and would partially dry. The heat of combustion of a wet cocoa pod husk is ~14 MJ/kg. A biomass to energy facility producing 1 MWe, and running at a 20% thermal efficiency, requires 5 MW of heat of combustion, or a supply of 1286 kg/h of wet cocoa pod husk. The annual requirement is ~8900 tons of dry cocoa pod husk. This requires a farming area of ~240 km$^2$, or a transportation radius of 17 km.

The implication of the thermodynamics is that the business model is better suited for multiple small facilities, rather than trying to build one very large production facility. The cocoa bean production in Cote d'Ivoire is ~2 million ton/annum, with an equivalent amount of dry cocoa pod husk produced. If it all could be used for electricity generation, the potential production capacity would be 225 MWe, which is roughly 10% of the current installed electrical capacity (2200 MWe). A small number of local producing sites will not impact the national electrical grid but will help in electrifying the remote farming communities that are not connected to the grid.

## 3.3. Project Scaling Issues

It is important to note that with an annual production of ~3 million ton of dry cocoa beans between Cote d'Ivoire and Ghana, there is sufficient cocoa pod husk available to saturate the global potassium carbonate market (~450,000 ton/annum). The difficulty is developing the infrastructure to collect all of the cocoa pod husks, to convert them into ash, and then to process the ash. The lack of transportation infrastructure and the low energy density of cocoa pod husk limit the scale of production that can be achieved.

The potassium carbonate product from the cocoa pod husk can readily displace some of the imported potassium carbonate. It also can be used to grow the production of palm oil-based soaps. These soaps are locally produced but not exported due to the requirement to import the alkali. The lower-cost biomass-derived alkali enables the production of soaps for export.

## 4. Materials and Methods

Samples of cocoa pod husk were obtained from farms in various growing regions of Cote d'Ivoire. The husk samples were analyzed using standard methods for % dry matter, % organic matter, and the elements C, N, P, K, Na, Ca, and Mg, along with higher and lower heating values. The ash produced

was analyzed for the elements Al, Ba, Ca, Cd, Cl, Co, Cr, Cu, F, Fe, K, Mg, Mn, Mo, Na, Ni, P, Pb, S, Si, Ti, V, and Zn.

The information from these analyses was compared with values in the open literature. Material and energy balance calculations were performed at various scales, using EXCEL as the calculation tool. Next, the material and energy balance calculations were balanced against a farm-scale consideration, with the issue being to determine the supply for a local farm community, and the electrical generation capacity that could be obtained for that community.

The results of the calculations were compared with results of pilot plant data, obtained from a 10s of kg/h pilot plant, operated by Organic Potash Ltd., in Tema, Ghana. The calculations presented herein discuss the broader socio-economic aspects of the project.

## 5. Conclusions

The production of potassium carbonate from biomass-derived ash is not a new process. Cocoa has a high potassium content, and this enables the use of cocoa biomass ash as a raw material for both fertilizer and for chemical production. Normally, one would compost the biomass as a means of improving soil quality; this would return the inorganic matter to the soil as well as returning organic matter. Unfortunately, the prevalence of viral diseases precludes the use of composting as a means of returning the cocoa biomass to the soil.

The use of cocoa biomass for electricity production allows for a circular and more sustainable economy to be developed. The remote farming villages become electrified through connection to a microgrid; the standard of living of the farmers is improved without the need to import fossil fuels. The areas where the cocoa biomass is currently left to rot are cleared, which expands the fraction of arable land. The farmers have additional sources of income due to the sale of the cocoa biomass. Additionally, the filter cake is provided as a low-cost calcium, magnesium, and phosphate fertilizer, which reduces the need to import calcium-based fertilizers. Further, a small potassium chemical business will be created; the business would allow for a reduction in the imports of potassium chemicals, as well as enabling the creation of a lower-cost soap export business. Further, this decentralized business approach could be applied to some of the cooperatives, the local community organizations that consolidate the cocoa from individual farmers.

**Author Contributions:** Conceptualization, K.A., G.N.; methodology, K.A., G.N.; validation, K.K., K.A., G.N.; formal analysis, G.N.; investigation, K.K., K.A., G.N.; resources, K.K., K.A., G.N.; data curation, G.N.; writing—original draft preparation, G.N.; writing—review and editing, K.K., K.A., G.N.; project administration, G.N. All authors have read and agreed to the published version of the manuscript.

**Funding:** This research received no external funding.

**Acknowledgments:** The authors acknowledge the support of Organic Potash Ltd. for granting permission to publish this article.

**Conflicts of Interest:** The authors declare no conflict of interest.

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
