# Peer review of "On the Production of Potassium Carbonate from Cocoa Pod Husks"

_recycling, doi:10.3390/recycling5030023_

Round 1

Reviewer 1 Report

in table 1, I would advise more  explanation of the results.  the last column should be titled per ha

line 189, the yield of dry pod is missing: 26.25 T

The potential of electricity production using 50% of total dry pod  in Ivory Coast  could be estimated  and compared to total electricity produced in Ivory Coast in 2019 (2200 MW) to demontrate the relevance of such option

The potential of Potassium fertilizer production could be highlighted knowing the Cocoa bean production in Ivory coast is around 2.036 million Tons of beans in 2019

I would suggest  to propose a decentralized capacity of processing electricity and producing eventually cocoa pod based K fertilizer  which could be adapted with selected cooperatives (over 1000 tons of bean collected)

Author Response

We have added a paragraph of explanation for Table 1, and adjusted the units; the yield of dry pod has been added (no longer line 189); the potential electrical production has been discussed with regards to Cote d’Ivoire’s grid capacity; a comment on the potential potassium fertilizer market has been added; and, we have made a note regarding the link to cooperatives.

Full editorial response letter attached.

Reviewer 2 Report

The manuscript reports the feasibility study of the potassium carbonate recovery from cocoa pod husks. The topic can have a relevant interest for the regions where cocoa is grown. Therefore, the study applicability is evident.

However, the present version shows several defects, that can be corrected with major revisions, in order to make clear the study and its results to the readers:

  1. the INTRODUCTION repeats information of the cocoa characteristics more times here and there (this defect occurs also in other part of the manuscript). These repetitions must be avoided;
  2. in the current version, the section MATERIALS AND METHODS does not report any information on what it was really carried out during the study, to say the content does not give any procedural/experimental information adopted by the Authors. In other terms, it is not clear if the authors carried out experimental runs or they simply used literature data to calculate the potentiality of the idea. The section must be improved and made clear to the readers;
  3. the section RESULTS must contain the results achieved during the study, and not descriptions suitable for INTRODUCTION (for example, a large parte of the section 2.2);
  4. the section DISCUSSION must refer to the achieved RESULTS, not simply show results here and there (for example, rows 187-196);
  5. the rows 203-213 are useless, and in any case not proper for the section DISCUSSION;
  6. row 189: the value of yield is missing;
  7. row 191: the reference is missing.

About editing, the Authors must be more careful to the INSTRUCTIONS FOR AUTHORS. Moreover, the institution no. 2 (Karl Akueson) is missing.

In general, I strongly recommend to do major revisions and transform the type of paper into COMMUNICATIONS.

Author Response

distinction between the sections of the articles;   In this regards – we have added 2 paragraphs of additional details to the Methods section (2); we have removed much of the test from the Results section (3 - some has been deleted and some reworked into Discussion); the discussion section is focussed in a more traditional fashion (4); we have removed the sentences on Dutching cocoa as noted (5); the yield has been added, and the reference is to the heating value now reported in Results (6 and 7); and the introduction has been adjusted in light of the above edits (1).  Further, we have added a corporate identity for Mr. Akueson (a private citizen, who doesn’t have an academic affiliation).

I note that Reviewer 2 commented about this submission being a “Communication”, rather than a “Research Paper”; we leave that to the editor to decide - we are content either way.

Full réponse letter attached

Round 2

Reviewer 2 Report

-